# BayReL: Bayesian Relational Learning for Multi-omics Data Integration

**Ehsan Hajiramezanali**\*, **Arman Hasanzadeh**\*, **Nick Duffield, Krishna Narayanan,**
**Xiaoning Qian**

Department of Electrical and Computer Engineering, Texas A&M University
`{ehsanr, armanihm, duffieldng, krn, xqian}@tamu.edu`

## Abstract

High-throughput molecular profiling technologies have produced high-dimensional multi-omics data, enabling systematic understanding of living systems at the genome scale. Studying molecular interactions across different data types helps reveal signal transduction mechanisms across different classes of molecules. In this paper, we develop a novel Bayesian representation learning method that infers the relational interactions across multi-omics data types. Our method, **Bay**esian **Re**lational **L**earning (BayReL) for multi-omics data integration, takes advantage of *a priori* known relationships among the same class of molecules, modeled as a graph at each corresponding view, to learn view-specific latent variables as well as a multi-partite graph that encodes the interactions across views. Our experiments on several real-world datasets demonstrate enhanced performance of BayReL in inferring meaningful interactions compared to existing baselines.

## 1 Introduction

Modern high-throughput molecular profiling technologies have produced rich high-dimensional data for different bio-molecules at the genome, constituting genome, transcriptome, translatome, proteome, metabolome, epigenome, and interactome scales [Huang et al., 2017, Hajiramezanali et al., 2018b, 2019b, Karimi et al., 2020, Pakbin et al., 2018]. Although such multi-view (multi-omics) data span a diverse range of cellular activities, developing an understanding of how these data types quantitatively relate to each other and to phenotypic characteristics remains elusive. Life and disease systems are highly non-linear, dynamic, and heterogeneous due to complex interactions not only within the same classes of molecules but also across different classes [Andrés-León et al., 2017]. One of the most important bioinformatics tasks when analyzing such multi-omics data is how we may integrate multiple data types for deriving better insights into the underlying biological mechanisms. Due to the heterogeneity and high-dimensional nature of multi-omics data, it is necessary to develop effective and affordable learning methods for their integration and analysis [Huang et al., 2017, Hajiramezanali et al., 2018a].

Modeling data across two views with the goal of extracting shared components has been typically performed by canonical correlation analysis (CCA). Given two random vectors, CCA aims to find the linear projections into a shared latent space for which the projected vectors are maximally correlated, which can help understand the overall dependency structure between these two random vectors [Thompson, 1984]. However, it is well known that the classical CCA suffers from a lack of probabilistic interpretation when applied to high dimensional data [Klami et al., 2013] and it also cannot handle non-linearity [Andrew et al., 2013]. To address these issues, probabilistic CCA (PCCA) has been proposed and extended to non-linear settings using kernel methods and neural networks

---

[Bach and Jordan, 2005]. Due to explicit uncertainty modeling, PCCA is particularly attractive for biomedical data of small sample sizes but high-dimensional features [Ghahramani, 2015, Huo et al., 2020].

Despite the success of the existing CCA methods, their main limitation is that they do not exploit structural information among features that is available for biological data such as gene-gene and protein-protein interactions when analyzing multi-omics data. Using available structural information, one can gain better understanding and obtain more biologically meaningful results. Besides that, traditional CCA methods focus on aggregated association across data but are often difficult to interpret and are not very effective for inferring interactions between individual features of different datasets.

The presented work contains three major contributions: 1) We propose a novel Bayesian relation learning framework, BayReL, that can flexibly incorporate the available graph dependency structure of each view. 2) It can exploit non-linear transformations and provide probabilistic interpretation simultaneously. 3) It can infer interactions across different heterogeneous features of input datasets, which is critical to derive meaningful biological knowledge for integrative multi-omics data analysis.

## 2   Method

We propose a new graph-structured data integration method, **Bay**esian **Re**lational **L**earning (BayReL), for integrative analysis of multi-omics data. Consider data for different molecular classes as corresponding data views. For each view, we are given a graph $G_v = (\mathcal{V}_v, \mathcal{E}_v)$ with $N_v = |\mathcal{V}_v|$ nodes, adjacency matrix $\mathbf{A}^v$, and node features in a $N_v \times D$ matrix $\mathbf{X}_v$. We note that $G_v$ is completely defined by $\mathbf{A}^v$, hence we use them interchangeably where it does not cause confusion. We define sets $\mathcal{G} = \{G_1, \ldots G_V\}$ and $\mathcal{X} = \{\mathbf{X}_1, \ldots \mathbf{X}_V\}$ as the input graphs and attributes of all $V$ views. The goal of our model is to find *inter-relations* between nodes of the graphs in different views. We model these relations as edges of a multi-partite graph $\mathfrak{G}$. The nodes in the multi-partite graph $\mathfrak{G}$ are the union of the nodes in all views, i.e. $\mathcal{V}_{\mathfrak{G}} = \bigcup_{v=1}^{V} \mathcal{V}_v$; and the edges, that will be inferred in our model, are captured in a multi-adjacency tensor $\mathcal{A} = \{\mathbf{A}^{vv'}\}_{v,v'=1,v\neq v'}^{V}$ where $\mathbf{A}^{vv'}$ is the $N_v \times N_{v'}$ bi-adjacency matrix between $\mathcal{V}_v$ and $\mathcal{V}_{v'}$. We emphasize that unlike matrix completion models, none of the edges in $\mathfrak{G}$ are assumed to be observed in our model. We infer our proposed probabilistic model using variational inference. We now introduce each of the involved latent variables in our model as well as their corresponding prior and posterior distributions. The graphical model of BayReL is illustrated in Figure 1.

**Embedding nodes to the latent space.** The first step is to embed the nodes in each view into a $D_u$ dimensional latent space. We use view-specific latent representations, denoted by a set of $N_v \times D_u$ matrices $\mathcal{U} = \{\mathbf{U}_v\}_{v=1}^{V}$, to reconstruct the graphs as well as inferring the inter-relations. In particular, we parametrize the distribution over the adjacency matrix of each view $\mathbf{A}^v$ independently:

$$\int p_\theta(\mathcal{G}, \mathcal{U}) \, d\mathcal{U} = \prod_{v=1}^{V} \int p_\theta(\mathbf{A}^v, \mathbf{U}_v) \, d\mathbf{U}_v = \prod_{v=1}^{V} \int p_\theta(\mathbf{A}^v \mid \mathbf{U}_v) \, p(\mathbf{U}_v) \, d\mathbf{U}_v, \tag{1}$$

where we employ standard diagonal Gaussian as the prior distribution for $\mathbf{U}_v$'s. Given the input data $\{\mathbf{X}_v, G_v\}_{v=1}^{V}$, we approximate the distribution of $\mathcal{U}$ with a factorized posterior distribution:

$$q(\mathcal{U} \mid \mathcal{X}, \mathcal{G}) = \prod_{v=1}^{V} q(\mathbf{U}_v \mid \mathbf{X}_v, G_v) = \prod_{v=1}^{V} \prod_{i=1}^{N_v} q(\mathbf{u}_{i,v} \mid \mathbf{X}_v, G_v), \tag{2}$$

where $q(\mathbf{u}_{i,v} \mid \mathbf{X}_v, G_v)$ can be any parametric or non-parametric distribution that is derived from the input data. For simplicity, we use diagonal Gaussian whose parameters are a function of the input. More specifically, we use two functions denoted by $\varphi_v^{\text{emb},\mu}(\mathbf{X}_v, G_v)$ and $\varphi_v^{\text{emb},\sigma}(\mathbf{X}_v, G_v)$ to infer the mean and variance of the posterior at each view from input data. These functions could be highly flexible functions that can capture graph structure such as many variants of graph neural networks including GCN [Defferrard et al., 2016, Kipf and Welling, 2017], GraphSAGE [Hamilton et al., 2017], and GIN [Xu et al., 2019]. We reconstruct the graphs at each view by deploying inner-product decoder on view specific latent representations. More specifically,

$$p(\mathcal{G} \mid \mathcal{U}) = \prod_{v=1}^{V} \prod_{i,j=1}^{N_v} p(\mathbf{A}_{ij}^v \mid \mathbf{u}_{i,v}, \mathbf{u}_{j,v}); \qquad p(\mathbf{A}_{ij}^v \mid \mathbf{u}_{i,v}, \mathbf{u}_{j,v}) = \text{Bernoulli}\left(\sigma(\mathbf{u}_{i,v} \, \mathbf{u}_{j,v}^T)\right), \tag{3}$$

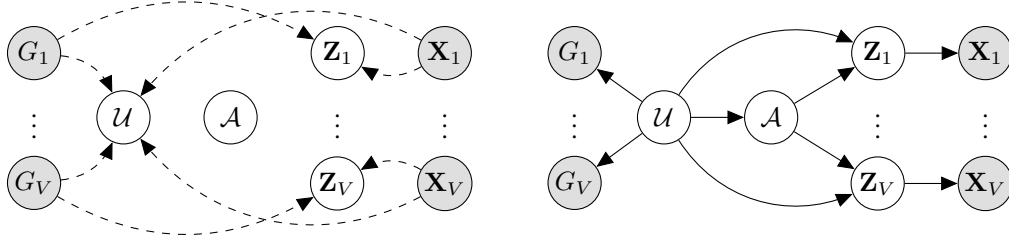

Figure 1: Graphical model for our proposed BayReL. **Left:** Inference; **Right:** Generative model.

where $\sigma(\cdot)$ is the sigmoid function. The above formulation for node embedding ensures that similar nodes at each view are close to each other in the latent space.

**Constructing relational multi-partite graph.** The next step is to construct a dependency graph among the nodes across different views. Given the latent embedding $\mathcal{U}$ that we obtain as described previously, we construct a set of bipartite graphs with multi-adjacency tensor $\mathcal{A} = \{\mathbf{A}^{vv'}\}_{v,v'=1,v\neq v'}^{V}$, where $\mathbf{A}^{vv'}$ is the bi-adjacency matrix between $\mathcal{V}_v$ and $\mathcal{V}_{v'}$. $\mathbf{A}_{ij}^{vv'} = 1$ if the node $i$ in view $v$ is connected to the node $j$ in view $v'$. We model the elements of these bi-adjacency matrices as Bernoulli random variables. More specifically, the distribution of bi-adjacency matrices are defined as follows

$$p(\mathbf{A}^{vv'} \mid \mathbf{U}_v, \mathbf{U}_{v'}) = \prod_{i=1}^{N_v}\prod_{j=1}^{N_{v'}} \text{Bernoulli}\left(\mathbf{A}_{ij}^{vv'} \mid \varphi^{\text{sim}}(\mathbf{u}_{i,v}, \mathbf{u}_{j,v'})\right), \qquad (4)$$

where $\varphi^{\text{sim}}(\cdot,\cdot)$ is a score function measuring the similarity between the latent representations of nodes. The inner-product link [Hajiramezanali et al., 2019a] decoder $\varphi_{\text{ip}}^{\text{sim}}(\mathbf{u}_{i,v}, \mathbf{u}_{j,v'}) = \sigma(\mathbf{u}_{i,v}\,\mathbf{u}_{j,v'}^T)$ and Bernoulli-Poisson link [Hasanzadeh et al., 2019] decoder $\varphi_{\text{bp}}^{\text{sim}}(\mathbf{u}_{i,v}, \mathbf{u}_{j,v'}) = 1 - \exp(-\sum_{k=1}^{D_u} \tau_k\, u_{ik,v}\, u_{jk,v'})$ are two examples of potential score functions. In practice, we use the concrete relaxation [Gal et al., 2017, Hasanzadeh et al., 2020] during training while we sample from Bernoulli distributions in the testing phase.

We should point out that in many cases, we have a hierarchical structure between views. For example, in systems biology, proteins are products of genes. In these scenarios, we can construct the set of directed bipartite graphs, where the direction of edges embeds the hierarchy between nodes in different views. We may use an asymmetric score function or prior knowledge to encode the direction of edges. We leave this for future study.

**Inferring view-specific latent variables.** Having obtained the node representations $\mathcal{U}$ and the dependency multi-adjacency tensor $\mathcal{A}$, we can construct view-specific latent variables, denoted by set of $N_v \times D_z$ matrices $\mathcal{Z} = \{\mathbf{Z}_v\}_{v=1}^{V}$, which can be used to reconstruct the input node attributes. We parametrize the distributions for node attributes at each view independently as follows

$$\int p_\theta(\mathcal{X}, \mathcal{Z} \mid \mathcal{G}, \mathcal{A}, \mathcal{U})\, \mathrm{d}\mathcal{Z} \;=\; \prod_{v=1}^{V}\prod_{i=1}^{N_v} \int p_\theta\left(\mathbf{z}_{i,v} \mid \mathcal{G}, \mathcal{A}, \mathcal{U}\right) p_\theta(\mathbf{x}_{i,v} \mid \mathbf{z}_{i,v})\, \mathrm{d}\mathbf{z}_{i,v}. \qquad (5)$$

In our formulation, the distribution of $\mathcal{X}$ is dependent on the graph structure at each view as well as inter-relations across views. This allows the local latent variable $\mathbf{z}_{i,v}$ to summarize the information from the neighboring nodes. We set the prior distribution over $\mathbf{z}_{i,v}$ as a diagonal Gaussian whose parameters are a function of $\mathbf{A}$ and $\mathcal{U}$. More specifically, first we construct the overall graph consisting of all the nodes and edges in all multi-partite graphs. We can view $\mathcal{U}$ as node attributes on this overall graph. We apply a graph neural network over this overall graph and its attributes to construct the prior. More formally, the following prior is adopted:

$$p_\theta(\mathcal{Z} \mid \mathcal{G}, \mathcal{A}, \mathcal{U}) = \prod_{v=1}^{V}\prod_{i=1}^{N_v} p_\theta(\mathbf{z}_{i,v} \mid \mathcal{G}, \mathcal{A}, \mathcal{U}), \qquad p_\theta(\mathbf{z}_{i,v} \mid \mathcal{G}, \mathcal{A}, \mathcal{U}) = \mathcal{N}(\boldsymbol{\mu}_{i,v}^{\text{prior}}, \boldsymbol{\sigma}_{i,v}^{\text{prior}}), \qquad (6)$$

where $\boldsymbol{\mu}^{\text{prior}} = [\boldsymbol{\mu}_{i,v}^{\text{prior}}]_{i,v} = \varphi^{\text{prior},\mu}(\mathcal{A}, \mathcal{U})$, $\boldsymbol{\sigma}^{\text{prior}} = [\boldsymbol{\sigma}_{i,v}^{\text{prior}}]_{i,v} = \varphi^{\text{prior},\sigma}(\mathcal{A}, \mathcal{U})$, and $\varphi^{\text{prior},\mu}$ and $\varphi^{\text{prior},\sigma}$ are graph neural networks. Given input $\{\mathbf{X}_v, G_v\}_{v=1}^{V}$, we approximate the posterior of

latent variables with the following variational distribution:

$$q(\mathcal{Z} \,|\, \mathcal{X}, \mathcal{G}) = \prod_{v=1}^{V} \prod_{i=1}^{N_v} q(\mathbf{z}_{i,v} \,|\, \mathbf{X}_v, G_v), \qquad q(\mathbf{z}_{i,v} \,|\, \mathbf{X}_v, G_v) = \mathcal{N}(\boldsymbol{\mu}_{i,v}^{\text{post}}, \boldsymbol{\sigma}_{i,v}^{\text{post}}) \qquad (7)$$

where $\boldsymbol{\mu}^{\text{post}} = [\boldsymbol{\mu}_{i,v}^{\text{post}}]_{i,v} = \{\varphi_v^{\text{post},\mu}(\mathbf{X}_v, G_v)\}_{v=1}^{V}$, $\boldsymbol{\sigma}^{\text{post}} = [\boldsymbol{\sigma}_{i,v}^{\text{post}}]_{i,v} = \{\varphi_v^{\text{post},\sigma}(\mathbf{X}_v, G_v)\}_{v=1}^{V}$, and $\varphi_v^{\text{post},\mu}$ and $\varphi_v^{\text{post},\sigma}$ are graph neural networks. The distribution over node attributes $p_\theta(\mathbf{x}_{i,v} \,|\, \mathbf{z}_{i,v})$ can vary based on the given data type. For instance, if $\mathcal{X}$ is count data it can be modeled by a Poisson distribution; if it is continuous, Gaussian may be an appropriate choice. In our experiments, we model the node attributes as normally distributed with a fixed variance, and we reconstruct the mean of the node attributes at each view by employing a fully connected neural network $\varphi_v^{\text{dec}}$ that operates on $\mathbf{z}_{i,v}$'s independently.

**Overall likelihood and learning.** Putting everything together, the marginal likelihood is

$$p_\theta(\mathcal{X}, \mathcal{G}) = \int \prod_{v=1}^{V} p_\theta(\mathbf{X}_v \,|\, \mathbf{Z}_v)\, p_\theta(\mathbf{Z}_v \,|\, \mathcal{G}, \mathcal{A}, \mathcal{U})\, p(\mathcal{A} \,|\, \mathcal{U})\, p(\mathcal{G} \,|\, \mathcal{U})\, p(\mathcal{U}) \, \mathrm{d}\mathbf{Z}_1 \ldots \mathrm{d}\mathbf{Z}_V \, \mathrm{d}\mathcal{A} \, \mathrm{d}\mathcal{U}.$$

We deploy variational inference to optimize the model parameters $\theta$ and variational parameters $\phi$ by minimizing the following derived Evidence Lower Bound (ELBO) for BayReL:

$$\mathcal{L} = \sum_{v=1}^{V} \Big[ \mathbb{E}_{q_\phi(\mathbf{Z}_v \,|\, \mathcal{G}, \mathcal{X})} \log p_\theta(\mathbf{X}_v \,|\, \mathbf{Z}_v) + \mathbb{E}_{q_\phi(\mathbf{Z}_v, \mathcal{U} \,|\, \mathcal{G}, \mathcal{X})} \log p_\theta(\mathbf{Z}_v \,|\, \mathcal{G}, \mathcal{A}, \mathcal{U})$$

$$- \mathbb{E}_{q_\phi(\mathbf{Z}_v \,|\, \mathcal{G}, \mathcal{X})} q_\phi(\mathbf{Z}_v \,|\, \mathcal{G}, \mathcal{X}) \Big] - \mathrm{KL}\left( q_\phi(\mathcal{U} \,|\, \mathcal{G}, \mathcal{X}) \,||\, p(\mathcal{U}) \right), \qquad (8)$$

where KL denotes the Kullback–Leibler divergence.

# 3 Related works

**Graph-regularized CCA (gCCA).** There are several recent CCA extensions that learn shared low-dimensional representations of multiple sources using the graph-induced knowledge of common sources [Chen et al., 2019, 2018]. They directly impose the dependency graph between *samples* into a regularizer term, but are not capable of considering the dependency graph between *features*. These methods are closely related to classic graph-aware regularizers for dimension reduction [Jiang et al., 2013], data reconstruction, clustering [Shang et al., 2012], and classification. Similar to classical CCA methods, they cannot cope with high-dimensional data of small sample sizes while multi-omics data is typically that way when studying complex disease. In addition, these methods focus on latent representation learning but do not explicitly model relational dependency between features across views. Hence, they often require ad-hoc post-processing steps, such as taking correlation and thresholding, to infer inter-relations.

**Bayesian CCA.** Beyond classical linear algebraic solution based CCA methods, there is a rich literature on generative modelling interpretation of CCA [Bach and Jordan, 2005, Virtanen et al., 2011, Klami et al., 2013]. These methods are attractive for their hierarchical construction, improving their interpretability and expressive power, as well as dealing with high dimensional data of small sample size [Argelaguet et al., 2018]. Some of them, such as [Bach and Jordan, 2005, Klami et al., 2013], are generic factor analysis models that decompose the data into shared and view-specific components and include an additional constraint to extract the statistical dependencies between views. Most of the generative methods retain the linear nature of CCA, but provide inference methods that are more robust than the classical solution. There are also a number of recent variational autoencoder based models that incorporate non-linearity in addition to having the probabilistic interpretability of CCA [Virtanen et al., 2011, Gundersen et al., 2019]. Our BayReL is similar as these methods in allowing non-linear transformations. However, these models attempt to learn low-dimensional latent variables for multiple views while the focus of BayReL is to take advantage of *a priori* known relationships among features of the same type, modeled as a graph at each corresponding view, to infer a multi-partite graph that encodes the interactions across views.

**Link prediction.** In recent years, several graph neural network architectures have been shown to be effective for link prediction by low-dimensional embedding [Hamilton et al., 2017, Kipf and Welling,

2016, Hasanzadeh et al., 2019]. The majority of these methods do not incorporate heterogeneous graphs, with multiple types of nodes and edges, or graphs with heterogeneous node attributes [Zhang et al., 2019]. In this paper, we have to deal with multiple types of nodes, edges, and attributes in multi-omics data integration. The node embedding of our model is closely related to the Variational Graph AutoEncoder (VGAE) introduced by Kipf and Welling [2016]. However, the original VGAE is designed for node embedding in a single homogeneous graph setting while in our model we learn node embedding for all views. Furthermore, our model can be used for prediction of missing edges in each specific view. BayReL can also be adopted for graph transfer learning between two heterogeneous views to improve the link prediction in each view instead of learning them separately. We leave this for future study.

**Geometric matrix completion.** There have been attempts to incorporate graph structure in matrix completion for recommender systems [Berg et al., 2017, Monti et al., 2017, Kalofolias et al., 2014, Ma et al., 2011, Hasanzadeh et al., 2019]. These methods take advantage of the known item-item and user-user relationships and their attributes to complete the user-item rating matrix. These methods either add a graph-based regularizer [Kalofolias et al., 2014, Ma et al., 2011], or use graph neural networks [Monti et al., 2017] in their analyses. Our method is closely related to the latter one. However, all of these methods assume that the matrix (i.e. inter-relations) is partially observed while we do not require such an assumption in BayReL, which is inherent advantage of formulating the problem as a generative model. In most of existing integrative multi-omics data analyses, there are no *a priori* known inter-relations.

## 4 Experiments

We test the performance of BayReL on capturing meaningful inter-relations across views on three real-world datasets[1]. We compare our model with three baselines, Spearman's Rank Correlation Analysis (SRCA) of raw datasets, Bayesian CCA (BCCA) [Klami et al., 2013], and Multi-Omics Factor Analysis (MOFA) [Argelaguet et al., 2018]. We note that mathematically, MOFA and BCCA are similar, except that MOFA has been extended with Bernoulli and Poisson likelihoods for discrete omics datasets. We also emphasize that deep Bayesian CCA models as well as deep latent variable models are not capable of inferring inter-relations across views (even with post-processing). Specifically, these models derive low-dimensional non-linear embedding of the input samples. However, in the applications of our interest, we focus on identifying interactions between nodes across views. From this perspective, only matrix factorization based methods can achieve the similar utility for which the factor loading parameters can be used for downstream interaction analysis across views. Hence, we consider only BCCA, a well-known matrix factorization method, but not other deep latent models for benchmarking with multi-omics data.

We implement our model in TensorFlow [Abadi et al., 2015]. For all datasets, we used the same architecture for BayReL as follows: Two-layer GCNs are used with a shared 16-dimensional first layer and separate 8-dimensional output layers as $\varphi_v^{\text{emb},\mu}$, and $\varphi_v^{\text{emb},\sigma}$. We use the same embedding function for all views. Inner-product decoder is used for $\varphi_v^{\text{sim}}$. Also, we employ a one-layer 8-dimensional GCN as $\varphi^{\text{prior}}$ to learn the mean of the prior. We set the variance of the prior to be one. We deploy view-specific two-layer fully connected neural networks (FCNNs) with 16 and 8 dimensional layers, followed by a two-layer GCN (16 and 8 dimensional layers) shared across views as $\varphi_v^{\text{post},\mu}$, and $\varphi_v^{\text{post},\sigma}$. Finally, we use a view-specific three-layer FCNN (8, input_dim, and input_dim dimensional layers) as $\varphi_v^{\text{dec}}$. ReLU activation functions are used. The model is trained with Adam optimizer. Also in our experiments, we multiply the term $\log p_\theta(\mathbf{Z}_v \mid \mathcal{G}, \mathcal{A}, \mathcal{U})$ in the objective function by a scalar $\alpha = 30$ during training in order to infer more accurate inter-relations. To have a fair comparison we choose the same latent dimension for BCCA as BayReL, i.e. 8. All of our results are averaged over four runs with different random seeds. More implementation details are included in the supplement.

### 4.1 Microbiome-metabolome interactions in cystic fibrosis

To validate whether BayReL can detect known microbe-metabolite interactions, we consider a study on the lung mucus microbiome of patients with Cystic Fibrosis (CF).

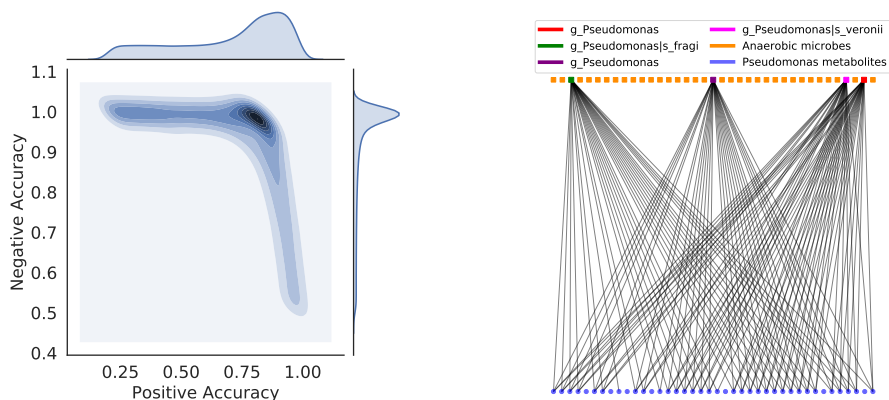

Figure 2: **Left**: Distribution of positive and negative accuracy in different training epochs for BayReL on CF dataset. **Right**: A sub-network of dependency graph consisting of *P. aeruginosa* micorbes, their validated targets, and anaerobic microbes, inferred using BayReL.

**Data description.** CF microbiome community within human lungs has been shown to be effected by altering the chemical environment [Quinn et al., 2015]. Anaerobes and pathogens, two major groups of microbes, dominate CF. While anaerobes dominate in low oxygen and low pH environments, pathogens, in particular *P. aeruginosa*, dominate in the opposite conditions [Morton et al., 2019]. The dataset includes 16S ribosomal RNA (rRNA) sequencing and metabolomics for 172 patients with CF. Following Morton et al. [2019], we filter out microbes that appear in less than ten samples, due to the overwhelming sparsity of microbiome data, resulting in 138 unique microbial taxa and 462 metabolite features. We use the reported target molecules of *P. aeruginosa* in studies Quinn et al. [2015] and Morton et al. [2019] as a validation set for the microbiome-metabolome interactions.

**Experimental details and evaluation metrics.** We first construct the microbiome and metabolomic networks based on their taxonomies and compound names, respectively. For the microbiome network, we perform a taxonomic enrichment analysis using Fisher's test and calculate p-values for each pairs of microbes. The Benjamini-Hochberg procedure [Benjamini and Hochberg, 1995] is adopted for multiple test correction and an edge is added between two microbes if the adjusted p-value is lower than 0.01, resulting in 984 edges in total. The graph density of the microbiome network is 0.102. For the metabolomics network, there are 1185 edges in total, with each edge representing a connection between metabolites via a same chemical construction [Morton et al., 2019]. The graph density of the metabolite network is 0.011.

We evaluate BayReL and baselines in two metrics – 1) accuracy to identify the validated molecules interacting with *P. aeruginosa* which will be referred as *positive accuracy*, 2) accuracy of *not* detecting common targets between anaerobic microbes and notable pathogen which we refer to this measure as *negative accuracy*. More specifically, we do not expect any common metabolite targets between known anaerobic microbes (*Veillonella*, *Fusobacterium*, *Prevotella*, and *Streptococcus*) and notable pathogen *P. aeruginosa*. If a metabolite molecule $x$ is associated with an anaerobic microbe $y$, then $x$ is more likely not to be associated with pathogen *P. aeruginosa* and vice versa. More formally, given two disjoint sets of metabolites $\mathfrak{s}_1$ and $\mathfrak{s}_2$ and the set of all microbes $\mathfrak{T}$ negative accuracy is defined as $1 - \frac{\sum_{i \in \mathfrak{s}_1} \sum_{j \in \mathfrak{s}_2} \sum_{k \in \mathfrak{T}} \mathbb{1}(i \text{ and } j \text{ are connected to } k)}{|\mathfrak{s}_1| \times |\mathfrak{s}_2| \times |\mathfrak{T}|}$, where $\mathbb{1}(\cdot)$ is an indicator function. Higher negative accuracy is better as there are fewer common targets between two sets of microbiomes.

**Numerical results.** Considering higher than $97\%$ negative accuracy, the best positive accuracy of BayReL, BCCA, MOFA, and SRCA are $82.7\% \pm 4.7$, $28.30\% \pm 3.21$, $28.13\% \pm 3.11$, and $26.41\%$, respectively. Clearly, BayReL substantially outperforms the baselines with up to $54\%$ margin. This shows that BayReL not only infers meaningful interactions with high accuracy, but also identify meicrobiome-metabolite pairs that should not interact. The performance of MOFA is very close to BCCA as expected. This is due to their similar mathematical formulations.

We also plot the distribution of positive and negative accuracy in different training epochs for BayReL (Figure 2). We can see that the mass is concentrated on the top right corner, indicating that BayReL consistently generates accurate interactions in the inferred bipartite graph. Figure 2 also shows

a sub-network of the inferred bipartite graph consisting *P. aeruginosa*, *anaerobic* microbes, and validated target nodes of *P. aeruginosa* and all of the inferred interactions by BayReL between them. While 78% of the validated edges of *P. aeruginosa* are identified by BayReL, it did not identify any connection between validated targets of *P. aeruginosa* and anaerobic microbes, i.e. negative accuracy of 100%. However, BCCA at negative accuracy of 100% could identify only one of these validated interactions. This clearly shows the effectiveness and interpretability of BayReL to identify inter-interactions.

When checking the top ten microbiome-metabolite interactions based on the predicted interaction probabilities, we find that four of them have been reported in other studies investigating CF. Among them, microbiome *Bifidobacterium*, marker of a healthy gut microbiota, has been qPCR validated to be less abundant in CF patients [Miragoli et al., 2017]. *Actinobacillus* and *capnocytophaga*, are commonly detected by molecular methods in CF respiratory secretions [Bevivino et al., 2019]. Significant decreases in the proportions of *Dialister* has been reported in CF patients receiving PPI therapy [Burke et al., 2017].

## 4.2 miRNA-mRNA interactions in breast cancer

We further validate whether BayReL can identify potential microRNA (miRNA)-mRNA interactions contributing to pathogenesis of breast cancer, by integrating miRNA expression with RNA sequencing (RNA-Seq) data from The Cancer Genome Atlas (TCGA) dataset [Tomczak et al., 2015].

**Data description.** It has been shown that miRNAs play critical roles in regulating genes in cell proliferation [Skok et al., 2019, Meng et al., 2015]. To identify miRNA-mRNA interactions that have a combined effect on a cancer pathogenesis, we conduct an integrative analysis of miRNA expressions with the consequential alteration of expression profiles in target mRNAs. The TCGA data contains both miRNA and gene expression data for 1156 breast cancer (BRCA) tumor patients. For RNA-Seq data, we filter out the genes with low expression, requiring each gene to have at least 10 count per million in at least 25% of the samples, resulting in 11872 genes for our analysis. We further remove the sequencing depth effect using edgeR [Robinson et al., 2010]. For miRNA data, we have the expression data of 432 miRNAs in total.

**Experimental details and evaluation metrics.** To take into account mRNA-mRNA and miRNA-miRNA interactions due to their involved roles in tumorigenesis, we construct a gene regulatory network (GRN) based on publicly available BRCA expression data from Clinical Proteomic Tumor Analysis Consortium (CP-TAC) using the R package GENIE3 [Vân Anh Huynh-Thu et al., 2010]. For the miRNA-

Table 1: Comparison of prediction sensitivity (in %) in TCGA for different graph densities.

| Avg. deg. | SRCA | BCCA | BayReL |
|---|---|---|---|
| 0.2 | 17.58 | $21.08 \pm 0.0$ | $\mathbf{34.06} \pm 2.5$ |
| 0.3 | 28.26 | $31.18 \pm 0.7$ | $\mathbf{47.46} \pm 2.6$ |
| 0.4 | 37.55 | $41.12 \pm 0.2$ | $\mathbf{59.50} \pm 3.0$ |

miRNA interaction networks, we construct a weighted network based on the functional similarity between pairs of miRNAs using MISIM v2.0 [Li et al., 2019]. We used miRNA-mRNA interactions reported by miRNet [Fan and Xia, 2018] as validation set. We calculate prediction sensitivity of interactions among validated ones while tracking the average density of the overall constructed graphs. We note that predicting meaningful interactions while inferring sparse graphs is more desirable as the interactions are generally sparse.

**Numerical results.** The results for prediction sensitivity (i.e. true positive rate) of BayReL and baselines with different average node degrees based on the interaction probabilities in the inferred bipartite graph are reported in Table 1. As we can see, BayReL outperforms baselines by a large margin in all settings. With the increasing average node degree (i.e. more dense bipartite graph), the improvement in sensitivity is more substantial for BayReL. For this dataset, the prediction sensitivity of MOFA (in %) is 22.09, 32.95, and 42.17 for the average degree 0.2, 0.3, and 0.4, respectively. It slightly outperforms BCCA due to better modeling of RNA-seq count data.

We also investigate the robustness of BayReL and BCCA to the number of training samples. Table 2 shows the prediction sensitivity of both models while using different percentage of samples to train the models. Using 50% of all the samples, while the average prediction sensitivity of BayReL reduces less than 2% in the worst case scenario (i.e. average node density 0.20), BCCA's performance degraded around 6%. This clearly shows the robustness of BayReL to the number of training samples.

In addition, we compare BayReL and BCCA in terms of consistency of identifying significant miRNA-mRNA interactions as well. We leave out 75% and 50% of all samples to infer the bipartite graphs, and then compare them with the identified miRNA-mRNA interactions using all of the samples. The KL divergence values between two inferred bipartite graphs for BayReL are $0.35$ and $0.32$ when using 25% and 50% of samples, respectively. The KL divergence values for BCCA are $0.67$ and $0.62$, using 25% and 50% of samples, respectively. The results prove that BayReL performs better than BCCA with fewer number of observed samples.

To further show the interpretability of BayReL, we inspect the top inferred interactions. Within them, multiple miRNAs appeared repeatedly. One of them is mir-155 which has been shown to regulate cell survival, growth, and chemosensitivity by targeting FOXO3 in breast cancer [Kong et al., 2010]. Another identified miRNA is mir-148b which has been reported as the biomarker for breast cancer prognosis [Shen et al., 2014].

### 4.3 Precision medicine in acute myeloid leukemia

We apply BayReL to identify molecular markers for targeted treatment of acute myeloid leukemia (AML) by integrating gene expression profiles and *in vitro* sensitivity of tumor samples to chemotherapy drugs with multi-omics prior information incorporated.

Classical multi-omics data integration to identify all gene markers of each drug faces several challenges. First, compared to the number of involved molecules and system complexity, the number of available samples for studying complex disease, such as cancer, is often limited, especially considering disease heterogeneity. Second, due to the many biological and experimental confounders, drug response could be associated with gene expressions that do not reflect the underlying drug's biological mechanism (i.e., false positive associations) [Barretina et al., 2012]. We show even with a small number of samples, BayReL improves the performance of the classical methods by incorporating prior knowledge.

**Data description.** This *in vitro* drug sensitivity study has both gene expression and drug sensitivity data to a panel of 160 chemotherapy drugs and targeted inhibitors across 30 AML patients [Lee et al., 2018]. While 62 drugs are approved by the U.S. Food and Drug Administration (FDA) and encompassed a broad range of drug action mechanisms, the others are investigational drugs for cancer patients. Following Lee et al. [2018], we study 53 out of 160 drugs that have less than 50% cell viability in at least half of the patient samples. Similar at the Cancer Cell Line Encyclopedia (CCLE) [Barretina et al., 2012] and MERGE [Lee et al., 2018] studies, we use the area under the curve (AUC) to indicate drug sensitivity across a range of drug concentrations. For gene expression, we pre-processed RNA-Seq data for 9073 genes [Lee et al., 2018].

**Experimental details and evaluation metrics.** To apply BayReL, we first construct the GRN based on the publicly available expression data of the 14 AML cell lines from CCLE using R package GENIE3. We also construct drug-drug interaction networks based on their action mechanisms. Specifically, the selected 53 drugs are categorized into 20 broad pharmacodynamics classes [Lee et al., 2018]; 14 classes contain more than one drugs. Only 16 out of the 53 drugs are shared across two classes. We consider that two drugs interact if they belong to the same class.

We evaluate BayReL on this dataset in two ways: 1) The prediction sensitivity of identifying reported drug-gene interactions based on 797 interactions archived in The Drug–Gene Interaction Database (DGIdb) [Wagner et al., 2016]. Note that DGIdb contains only the interactions for 43 of the 53 drugs included in our study. 2) Consistency of significant gene-drug interactions in two different AML datasets with 30 patients and 14 cell lines. We compare BayReL with BCCA in consistency of significant gene-drug interactions, where all 30 patient samples are used for discovery and the discovered interactions are validated using 14 cell lines.

Table 2: Prediction sensitivity (in %) in TCGA for different percentage of training samples.

| Avg. degree | BCCA | | | BayReL | | |
|---|---|---|---|---|---|---|
| | # of training samples | | | # of training samples | | |
| | 289 (25%) | 578 (50%) | 1156 (100%) | 289 (25%) | 578 (50%) | 1156 (100%) |
| 0.20 | $17.4 \pm 0.8$ | $17.6 \pm 1.0$ | $21.0 \pm 0.0$ | $31.9 \pm 3.0$ | $32.1 \pm 1.0$ | $34.0 \pm 2.5$ |
| 0.30 | $26.0 \pm 0.8$ | $26.4 \pm 1.0$ | $31.1 \pm 0.7$ | $45.8 \pm 3.1$ | $45.9 \pm 1.5$ | $47.4 \pm 2.6$ |
| 0.40 | $35.4 \pm 0.8$ | $35.5 \pm 0.7$ | $41.1 \pm 0.2$ | $57.6 \pm 4.4$ | $58.7 \pm 1.3$ | $59.5 \pm 3.0$ |

Table 3: Comparison of prediction sensitivity (in %) in AML dataset for different graph densities.

| Avg. degree | 0.10 | 0.15 | 0.20 | 0.25 | 0.30 | 0.40 | 0.50 |
|---|---|---|---|---|---|---|---|
| SRCA | 8.03 | 12.00 | 17.15 | 20.70 | 26.85 | 34.93 | 45.79 |
| BCCA | $9.65 \pm 0.75$ | $14.34 \pm 0.06$ | $18.96 \pm 0.42$ | $23.29 \pm 0.52$ | $28.22 \pm 0.66$ | $38.02 \pm 2.15$ | $46.88 \pm 1.88$ |
| BayReL | $\mathbf{15.56} \pm 0.75$ | $\mathbf{21.70} \pm 0.65$ | $\mathbf{27.20} \pm 0.17$ | $\mathbf{32.43} \pm 1.02$ | $\mathbf{37.76} \pm 0.85$ | $\mathbf{47.90} \pm 0.43$ | $\mathbf{56.76} \pm 0.50$ |

**Numerical results.** Table 3 shows BayReL outperforms both SRCA and BCCA at different average node degrees in terms of identifying validated gene-drug interactions. If we compare the results by BayReL and BCCA, their performance difference increases with the increasing density of the bipartite graph. While BayReL outperforms BCCA by 8% at the average degree 0.10, the improved margin increases to 10.7% at the average degree 0.50. This confirms that BayReL can identify potential gene-drug interactions more robustly.

We also compare the gene-drug interactions when we learn the graph using all 30 patient samples and 14 cell lines. The KL divergence between two inferred bipartite graphs are 0.38 and 0.66 for BayReL and BCCA, respectively. This could potentially account for the lower consistency rate of BCCA compared to BayReL. The capability of flexibly incorporating prior knowledge as view-specific graphs is an important factor for BayReL achieving more consistent results.

## 5    Conclusions

We have proposed BayReL, a novel Bayesian relational representation learning method that infers interactions across multi-omics data types. BayReL takes advantage of *a priori* known relationships among the same class of molecules, modeled as a graph at each corresponding view. By learning view-specific latent variables as well as a multi-partite graph, more accurate and robust interaction identification across views can be achieved. We have tested BayReL on three different real-world omics datasets, which demonstrates that not only BayReL captures meaningful inter-relations across views, but also it substantially outperforms competing methods in terms of prediction sensitivity, robustness, and consistency.

## Broader Impact

Our BayReL provides a general graph learning framework that can flexibly integrate prior knowledge when facing a limited number of training samples, which is often the case in scientific and biomedical applications. BayReL is unique in its model and potential applications. This novel generative model is able to deal with growing complexity and heterogeneity of modern large-scale data with complex dependency structures, which is especially critical when analyzing multi-omics data to derive biological insights, the main focus of our research.

Furthermore, learning with biomedical data can have significant impact in helping decision making in healthcare. However, decision making in biomedicine has to be robust and aware of potential data and prediction uncertainty as it can lead to significant consequences (life vs. death). Therefore, it is critical to develop accurate and reproducible results from new machine learning efforts. BayReL is a generative model with Bayesian modeling and robust variational inference and hence is equipped with natural uncertainty estimates, which will help derive reproducible and accurate prediction for robust decision making, with the ultimate goal of improving human health outcomes, as showcased in the three experiments.

## Acknowledgments and Disclosure of Funding

The presented materials are based upon the work supported by the National Science Foundation under Grants IIS-1848596, CCF-1553281, IIS-1812641, ECCS-1839816, and CCF-1934904.

## Footnotes

[1]Our code is available at `https://github.com/ehsanhajiramezanali/BayReL`

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
