[Supplementary Material]

# BayReL: Bayesian Relational Learning for Multi-omics Data Integration: Supplementary Materials

**Ehsan Hajiramezanali**\*, **Arman Hasanzadeh**\*, **Nick Duffield, Krishna Narayanan,**
**Xiaoning Qian**

Department of Electrical and Computer Engineering, Texas A&M University

## A. BayReL model

To further clarify the model and workflow of our proposed BayReL, we provide a schematic illustration of BayReL in Figure S1, where we only include two views for clarity. We again note that the framework and training/inference algorithms can be generalized to multiple ($> 2$) views though we have focused on the examples with two views due to the availability of the corresponding data and validation sets.

## B. Additional experimental results for acute myeloid leukemia (AML) data

Figure S2 shows the inferred bipartite network with the top 200 interactions by BayReL. Among the genes involved in these identified interactions, Secreted Phosphoprotein 1 (SPP1) has been considered as a prognostic marker of AML patients [4] for their sensitivity to different AML drugs. CD163 has been reported to be over-expressed in AML cells [1]. BayReL in fact has also identified corresponding drugs that have been proposed to target this gene. In addition, AML has been studied with the evidence that dysregulation of several pathways, including down-regulation of major histocompatibility complex (MHC) class II genes, such as human leukocyte antigen (HLA)-DPA1 and HLA-DQA1, involved in antigen presentation, may change immune functions influencing AML development [3]. BayReL has identified the interactions between these genes and some of the validated AML drugs as appropriate therapeutics to reverse the corresponding epigenetic changes.

To further demonstrate the biological relevance of the inferred drug-gene interactions by BayReL, gene ontology (GO) enrichment analysis with the genes among the top 200 interactions has been performed using Fisher's exact test. Table S1 shows that the enriched GO terms by these genes agree with the reported mechanistic understanding of AML disease development. The most significantly enriched GO terms include the MHC class II receptor activity ($p$-value $= 0.00099$), chemokine activity ($p$-value $= 0.00175$), MHC class II protein complex ($p$-value $= 0.00273$), chemokine receptor binding ($p$-value $= 0.00404$), and regulation of leukocyte tethering or rolling ($p$-value $= 0.00030$). The top molecular function (MF) GO term, the MHC class II receptor activity, encoded by human leukocyte antigen (HLA) class II genes, plays important roles in antigen presentation and initiation of immune responses [2].

For comparison, we have performed the same GO enrichment analysis for the top 200 drug-gene interactions inferred by BCCA. The most significantly enriched GO terms are telomere cap complex ($p$-value $= 0.00016$), nuclear telomere cap complex ($p$-value $= 0.00016$), positive regulation of telomere maintenance ($p$-value $= 0.00041$), telomeric DNA binding ($p$-value $= 0.00044$), and SRP-dependent cotranslational protein targeting to membrane ($p$-value $= 0.00058$). To the best of our knowledge, these enriched GO terms are not directly related to AML disease mechanisms.

---

Figure S1. Schematic illustration of BayReL.

Figure S2: The bipartite sub-network with the top 200 interactions inferred by BayReL in AML data, where only the top six genes and their associated drugs are labelled in the figure for better visualization. Genes and drugs are shown as blue and red nodes, respectively.

Table S1: Enriched GO terms for the top 200 interactions in AML data.

| GO ID | GO class | Description | p-value |
|---|---|---|---|
| GO:0006084 | BP | acetyl-CoA metabolic process | 0.00051 |
| GO:0009404 | BP | toxin metabolic process | 0.00099 |
| GO:0032395 | MF | MHC class II receptor activity | 0.00099 |
| GO:0033004 | BP | negative regulation of mast cell activation | 0.00099 |
| GO:0048240 | BP | sperm capacitation | 0.00099 |
| GO:0008009 | MF | chemokine activity | 0.00175 |
| GO:0042613 | CC | MHC class II protein complex | 0.00273 |
| GO:0042379 | MF | chemokine receptor binding | 0.00404 |

## C.  Negative accuracy threshold for cystic fibrosis (CF) data

While we only discussed the results for one specified threshold value for negative accuracy ($97\%$) in the paper for brevity, we here provide additional results with other threshold values, which show similar improvements by BayReL (see Figure S3). We note that there is a trade-off between positive and negative accuracies, and the optimal point can be chosen depending on the application.

Figure S3: Positive vs negative accuracy in CF data.

## D.  Details on the experimental setups, hyper-parameter selection, and run time

**BayReL.**    In all cystic fibrosis (CF) and acute myeloid leukemia (AML) experiments, the learning rate is set to be $0.01$. For the TCGA breast cancer (BRCA) dataset, the learning rates are $0.0005$ when using all and half of samples and $0.005$ when using 25% of all samples. We learn the model for 1000 training epochs and use the validation set for early stopping. All of the experiments are run on a single GPU node GeForce RTX 2080. Each training epoch for CF, BRCA, and AML took 0.01, 0.42, and 0.23 seconds, respectively.

**BCCA.**    In all experiments, we used CCAGFA R package as the official implementation of BCCA. We got the default hyper-parameters using the function 'getDefaultOpts' as suggested by the authors. We then construct the bi-partite graph using Spearman's rank correlation between the mean projections of views. We report the results based on four independent runs.