[Reviews · NeurIPS 2020]

Review 1

Summary and Contributions: This study proposes a Bayesian formulation for multiomics data integration by combining within-view and between-view interactions.

Strengths: The problem the authors addressed is a very important problem in computational biology. Their formulation seems novel.

Weaknesses: The \alpha parameter mentioned in line 180 were not mentioned before. This parameter basically determines the trade-off between within-view and between-view interactions. There is no discussion about the effect of this parameter on the solution quality. They compared the proposed algorithm mainly against BCCA algorithm, which is by definition using within-view interactions only. If otherwise, the authors should explain how they integrated between-view interactions into BCCA. I would suggest them to compare their algorithm against BCCA under the setting \alpha = 0 if within-view interactions are not used by BCCA.

Correctness: The claims and method seem correct.

Clarity: The paper is well written.

Relation to Prior Work: Most of the related prior work have been discussed. However, there are some co-embedding algorithms that makes use of within-view and between-view interactions, and they were not mentioned.

Reproducibility: Yes

Additional Feedback:


Review 2

Summary and Contributions: This work considers the problem of relation learning across multiple views where each view corresponds to an omic data type. The authors propose a Bayesian model for relation learning that accounts for known relations in each view and can leverage non-linear transformations. Inference in this model is performed using a variational formalism. The authors apply their model to 3 specific applications to demonstrate promising results.

Strengths: This work provides a coherent probabilistic framework that generalizes a number of previous studies. The full probabilistic treatment coupled with the flexibility is a strength. The empirical results across a range of applications are careful and convincing and support the broad utility of this model.

Weaknesses: A discussion of the computational requirements of inference and learning (relative to the baselines) would be important to add.

Correctness: - The empirical results are convincing. - While the comparison to BCCA and Spearman rank correlation are useful, it would be informative to probe the BayRel model directly to understand which of its components contribute to the improvements e.g. how important are the graphs vs the feature embeddings? Comparing BayRel to simpler sub-models could provide additional insights.

Clarity: Yes

Relation to Prior Work: Yes

Reproducibility: Yes

Additional Feedback: Update after reading author comments: I appreciate the additional results comparing BayRel to MOFA and the ablation experiments. This work presents a useful methodological contribution.


Review 3

Summary and Contributions: The authors present BayRel, which is a method for analysing linked data-sets from different genomics platforms. These linked data-sets may relate to different types of genomic data, such as gene-expression / transcriptomic data, or gene-regulatory data such as chromatin accessibility data. The main idea of BayRel is to uncover latent variables that give rise to the structure observed in the various data-sets (similar to canonical correlation analysis). In this context, these latent variables represent biological processes related to gene regulation that give rise to the patterns observed in the various types of genomics data.

Strengths: The problem of simultaneously analysing several types of genomics data that are linked is timely in the fields of cell biology and computational statistics. To approach this problem, these authors fit a sophisticated hierarchical probabilistic model using variational inference. The mathematical arguments are detailed and clear, the results are well presented.

Weaknesses: Conceptually, BayRel is very similar to Mult-Omics Factor Analysis (MOFA): https://doi.org/10.15252/msb.20178124 which also uses variational inference to fit a latent variable model simultaneously to multiple genomic data-sets, to infer common structure and variation corresponding to biological processes of interest. As MOFA is well established, has an equivalent level of mathematical and computational sophistication to BayRel, and has been validated in high-profile studies in the genomics application-area, I recommend that the authors benchmark BayRel against MOFA. Furthermore, I recommend that the authors specifically include the most important types of genomic data in this comparison: in particular, I think that they should include RNA-Seq and ChIP-Seq data. This would give a clearer picture of the relative merits of BayRel.

Correctness: The authors evaluate BayRel using some unusual types of genomic data: specifically, microbiome and micro-RNA data. As BayRel is designed to simultaneously analyse different types of genomic data in the most general sense, it seems important to first benchmark BayRel against much more common types of genomic data. Specifically, I would recommend the authors benchmark BayRel against gene-expression (such as RNA-seq) data in conjunction with ChIP-seq data as a minimum, and also including DNA methylation data if possible, before considering micro-RNA and microbiome data.

Clarity: The mathematical arguments are detailed and clear, the results are well presented, and the manuscript is easy to read.

Relation to Prior Work: Conceptually, BayRel is very similar to Mult-Omics Factor Analysis (MOFA): https://doi.org/10.15252/msb.20178124 This previous work was not mentioned in the current manuscript, but should be discussed.

Reproducibility: Yes

Additional Feedback:


Review 4

Summary and Contributions: In this paper, the authors propose a Bayesian representation learning framework that can infer links between heterogeneous graphs generated from multi-omics datasets. The main idea is to use the underlying relationship information within each dataset (or view) by modeling it as a graph. The method has 4 steps - (1) to embed the nodes of each view-specific graph into in the same latent space (2) generate a multi-view adjacency tensor using the similarity scores for node embeddings across views (3) Infer prior latent variables from the node embeddings and multi-view graphs and posterior from the view-specific data (4) Finally, perform variational inference to optimize model parameters and variational parameters. The paper attempts to solve an important problem of multi-omics data integration by learning relationships that can exist between different modalities by modeling them as multi-view link prediction. This work could be useful to the broader ML community.

Strengths: — The paper introduces an interesting Bayesian framework to perform multi-view link predictions by using the relationships within each view as prior information (in the form of graphs) -- The use of GCNs to learn node embeddings allows the method to capture the complex non-linear interactions in the data -- The probabilistic framework lends the method interpretation properties, useful for datasets in biology and clinical domain — This task is important for multi-domain data integration — The paper uses datasets from biology like (microbiome networks, gene, and drug interaction networks, etc.) to demonstrate that the proposed method outperforms the chosen baselines - Spearman’s Rank Correlation Analysis and Bayesian CCA - by producing higher precision scores — The paper also uses the scores for inferred interactions to verify the top interactions with evidence in the literature.

Weaknesses: — It would be useful to know how well the method scales to the size of the graphs for practical implementations — The paper mentions two different similarity functions (lines 83-84), which of these were used for the final results? The paper should clarify how the performance varies with this choice. 
— It is unclear why the scalar multiplication is useful (lines 180-181). How does this quantity affect the performance?

Correctness: It would be great if the following were clarified: — It is unclear if proper cross-validation was performed to select the final set of hyperparameters (for the method and the baselines). If not then the results could be misleading. — Is there a reason to just report the precision scores for the evaluation instead of false positives (important for clinical/biological tasks) or F1-scores? — Line 217, why the threshold for negative accuracy chosen to be 97% 
— Why is table 2 missing the SRCA baseline? — It would be useful if the results in lines 278-279 are quantified. For example, what was the % of the occurrence of mir-155 to be investigated by the authors? Was there a threshold to pick these examples?

Clarity: The paper is quite well written

Relation to Prior Work: Yes

Reproducibility: Yes

Additional Feedback: -- Figure 2 panel A: the negative accuracy axis has label 1.1 -- 
The broader impact section could be revised to be more specific towards applications as well as mention the potential issues/advantages of deploying such methods for the clinical domain. -- It is a little unclear why the graph-based neural networks (handling multi-view or heterogeneous data) could not be used as baselines for the paper

[Author Response · NeurIPS 2020]

We truly appreciate helpful comments from all the reviewers. We will add suggested references and try our best to
improve the presentation. We address the major ones here and would like first to emphasize that the main differences
of BayReL from existing omics integrative analysis methods (including MOFA) are: 1) it infers relations in a unified
formulation while the existing ones focus on integrating different data types to derive generic latent representations for
downstream bioinformatics tasks; 2) it incorporates the graph structure at each view which is crucial for performance
(see our response to **R2** below); 3) it learns the relations through non-linear/deep transformations of data as opposed to
linear ones in most of the existing methods; 4) unlike co-embedding and matrix completion based methods, it infers
relations between different molecular classes, **without any pre-known interactions across classes/views**.

[**R1** & **R4**] $\alpha$ in loss function is known as tempering [Huang, NeurIPS2018] in the context of Bayesian inference, which
helps smoothing out the objective function by promoting modes that are close to those of the prior and is designed for
better inference. We cannot set $\alpha$ to zero as it would remove the log prior term in our generative model. Appropriately
tuned $\alpha > 1$ outperforms the original VAE training ($\alpha = 1$). Empirically on the CF dataset, the positive accuracy (PA,
in %) of BayReL at 97% negative accuracy (NA) is 80.3, 82.3, 82.6, 82.7, and 81.2, with $\alpha = 1, 10, 20, 30$, and 50,
respectively. We emphasize that, in BayReL, none of the between-view interactions are assumed to be *a priori* known
in training. Changing the formulation to avoid using *learned* between-view interactions in prior construction, would
change the model and the overall likelihood substantially, which we leave for future studies.

[**R3**] The extension of BCCA (only considering 2 views) to multiple views is known as Bayesian group factor analysis
(GFA). In our experiments, we used the GFA implementation released in CCAGFA R package; but since we only
had 2-view datasets, we referred to it as BCCA to avoid confusion. We should point out that mathematically, MOFA
and GFA are similar, except that MOFA has been extended with Bernoulli and Poisson likelihoods for discrete omics
datasets. We evaluate MOFA on the CF dataset, its PA, while NA is set to 97%, is 28.13%, very close to BCCA as we
expected. For miRNA-mRNA, the prediction sensitivity of MOFA (in %) is 22.09, 32.95, & 42.17 for avg. deg. 0.2,
0.3, & 0.4, respectively. It slightly outperforms BCCA due to better modeling of RNA-seq count data.

[**R3**] The datasets we used have been extensively studied in the literature (e.g. MOFA studied precision medicine
and multi-modal microbiome) and as pointed out in the paper (lines 239-241 & 283), they are of great importance in
biology. We tried to pick three very different applications to show applicability of BayReL in biomedicine. Furthermore,
these datasets have two types of heterogeneity: both graph and node attributes are different across views. The
heterogeneity in the suggested RNA-seq/ChIP-seq would be only because of different types of node attributes (due to
different technologies used for characterizing properties of genes/proteins). We also note that miRNA, mRNA, and
gene-expression data in our 2nd and 3rd experiments, are indeed RNA-seq data. BayReL will be carefully evaluated
(considering practical challenges to validate cross-view relations) with multi-view heterogeneous omics data in future.

[**R2**] We tested two modified versions of BayReL: 1) to show the importance of reconstructing graph structures, we
removed the view-specific graph reconstruction (Equation 3) from the model (BayReL-NoRecon); 2) to show the
importance of using view-specific graphs, we assume view-specific adjacency matrices are identity (BayReL-NoGraph).
Applying to CF, BayReL-NoRecon has shown to be unstable in training with respect to random initialization (at 97%
NA, PA can be as low as 10% for some seeds and as high as 79% for others). We argue that the reason for such a
behavior is that removing graph reconstruction would cause the embedding to rotate arbitrarily with respect to views,
which leads to poor performance. Adding view-specific graph reconstruction ensures that node embeddings are faithful
to the view-specific graph structures as well as avoiding arbitrary rotations across views. BayReL-NoGraph also
performs worse than BayReL on CF with PA of 44.7% at 97% NA. We note that BayReL-NoGraph outperforms BCCA.

[**R4**] Negative accuracy threshold: While we only discussed the results for one PA thresh-
old (97%) in the paper for brevity, we can see similar improvements in other thresholds
too (see the figure on the right for CF data). We note that there is a trade-off between
PA and NA, and the optimal point is chosen based on the application. [**R4**] SRCA in
Table 2: We did not include it in the original submission for better layout and readability.
Prediction sensitivity of SRCA (in %) in TCGA for 25% and 50% of training samples
are 25.53 (33.75) & 27.10 (35.79), respectively, for avg. deg. 0.3 (0.4). We will include
a complete table in the supplement. Regarding computational complexity [**R2** & **R4**]:

We have reported the run time of BayReL in the supplement (Section C). Our current
implementation is relatively fast on mid-size graphs (~15K nodes, <0.5 second per epoch on a single GPU node),
indicating its scalability (with respect to the number of nodes, edges, and node attributes) to large datasets. In addition,
by deploying sampling based GCNs, e.g. GraphSAGE, FastGCN, and FastGAE, the scalability can be further improved.
Regarding accuracy measures [**R4**]: We haven't used F1 score because no true negatives are known in the datasets (only
some true positives are known, typical in bioinformatics). Regarding $\varphi^{\mathrm{sim}}$ [**R4**]: We used an inner-product decoder
(line 174 in the paper). Regarding hyperparameter selection [**R4**]: For BayReL, as most of the variational unsupervised
methods, we chose them based on the training cost. For BCCA, it was described in the supplement (Section C).

[Meta-Review · NeurIPS 2020]

The paper proposes a Bayesian formulation for the integration of multi omics datasets by combining within-view and between-view interactions. Although the paper is conceptually related to prior work, the reviewers appreciate the contributions made, which are both timely and relevant to the neurips community. Overall, this is a solid submission and the authors defend the concerns raised convincingly in their rebuttal.